# Mapping the energy landscapes of supramolecular assembly by thermal hysteresis

Robert W. Harkness, V[1], Nicole Avakyan[1], Hanadi F. Sleiman [1] & Anthony K. Mittermaier[1]

Understanding how biological macromolecules assemble into higher-order structures is critical to explaining their function in living organisms and engineered biomaterials. Transient, partly-structured intermediates are essential in many assembly processes and pathway selection, but are challenging to characterize. Here we present a simple thermal hysteresis method based on rapid, non-equilibrium melting and annealing measurements that maps the rate of supramolecular assembly as a function of temperature and concentration. A straightforward analysis of these surfaces provides detailed information on the natures of assembly pathways, offering temperature resolution beyond that accessible with conventional techniques. Validating the approach using a tetrameric guanine quadruplex, we obtain strikingly good agreement with previous kinetics measurements and reveal temperature-dependent changes to the assembly pathway. In an application to the recently discovered co-assembly of polydeoxyadenosine (poly(A)) and cyanuric acid, we show that fiber elongation is initiated when an unstable complex containing three poly(A) monomers acquires a fourth strand.

[1] Department of Chemistry, McGill University, 801 Sherbrooke St. W., Montreal Quebec H3A 0B8, Canada. Correspondence and requests for materials should be addressed to A.K.M. (email: anthony.mittermaier@mcgill.ca)

The non-covalent assembly of biological or biomimetic subunits into large supramolecular structures is critical to the function of living organisms and the creation of novel biomaterials. Supramolecular assemblies are validated drug targets[1,2], and contain tightly controlled internal structure on the nanometer scale, offering new opportunities in the bottom-up design of functional materials[3,4]. In general, these large supramolecular structures are too complicated to form in a single step, and instead follow multi-step assembly pathways comprising multiple transient, partly-assembled, intermediate states. The nature of these intermediates and the factors governing their interconversion are critical to understanding biological supramolecular self-assembly and yet remain poorly understood[5,6]. Assembly intermediates are often unstable and short-lived, thus direct detection is not always possible. Nevertheless, detailed information on assembly pathways and the intermediate states that comprise them can be obtained through careful study of how the overall reaction rate varies with environmental conditions, particularly monomer concentration, $[M]$, and temperature. In particular, for homomeric assembly, an effective reaction order, $n$, implies that the reaction rate varies as $[M]^n$, and can be related to the molecularity of the rate-determining transition state[7], or the critical nucleus size for polymerization[8,9], depending on the system. We find that measuring the effective reaction order as a function of temperature allows one to map the energy landscapes of supramolecular assembly with remarkable detail.

Measurements of self-assembly kinetics typically involve triggering the reaction by rapid mixing[10], temperature jump[11], or flash photolysis[12] and monitoring the accumulation of the assembled product as a function of time, while the temperature is held constant. In order to obtain robust measurements of the reaction order, these experiments must be repeated for different initial values of $[M]$[7]. This series of experiments must then be repeated for each temperature of interest. The costs in time and material are high for this type of analysis, which consequently has been performed on just a handful of systems with only modest temperature sampling[7]. Thermal hysteresis (TH) is a simple and rapid technique that has previously been used mainly to measure two-state folding and unfolding rates of biomolecules[13,14], as well as the magnetic properties[15,16] of inorganic nanoparticles. This experiment entails measuring a spectroscopic signature of folding or assembly (such as absorbance or ellipticity) while raising and lowering the temperature to cause melting and annealing. The temperature scan rate is chosen to be rapid compared to the length of time needed for the system to relax to equilibrium, such that both folding and unfolding occur out of equilibrium. The populations effectively lag behind the rapidly changing temperature such that the 50% folding point (i.e., the apparent melting temperature ($T_m$)) is reached at a higher temperature than the true $T_m$ on the up-scan and at a lower temperature than the true $T_m$ on the down-scan. The folding and unfolding rates can then be calculated as a function of temperature based on the size of the lag[13]. TH has been widely used to measure unimolecular folding and unfolding[17]. A small number of TH studies have examined multimeric assembly quantitatively, specifically trimer[18] and dimer[19] formation using explicit models of association, but not higher-order polymerization reactions.

Here we show that TH experiments have a great and largely untapped potential for de novo elucidation of complex supramolecular assembly pathways. In our method, data obtained from several (six in our study) different scan rates are combined to create a 3D map of reaction rate as a function of both $[M]$ and temperature. These surfaces are then analyzed in a two-step procedure. In the first step, effective assembly and disassembly reaction orders ($n$ and $m$, respectively) are extracted as a function of temperature in a model-free manner. A single set of experiments yields reaction orders across the entire thermal transition, spanning in our case up to 40 degrees, sampled in increments of 0.5 degrees, delivering a level of kinetic detail that is unattainable by conventional methods. In the second step, explicit mechanistic models are constructed, based on the observed reaction orders, and are globally fit to the TH datasets, simultaneously yielding the kinetic and thermodynamic parameters that quantify the supramolecular assembly pathway in terms of interconversion between partly-assembled intermediates. The combined model-free and global fitting approach can be applied to self-assembling systems that follow widely divergent mechanisms, as illustrated below.

## Results

**Model-free analysis of TH profiles**. Our model-free analysis is based on measuring sets of thermal melting and annealing spectrophotometric profiles with different temperature scan rates. The datasets contain low (assembled) and high (disassembled) temperature baselines bridged by transition regions where assembly and disassembly occur. Faster heating rates push the melting curves to successively higher temperatures as the populations lag further behind their equilibrium values. Conversely, faster cooling rates push the reannealing curves to successively lower temperatures. Each trace is then used to estimate the fraction of subunits that are dissociated (monomeric), $\theta_U$, as a function of temperature and scan rate (see Supplementary Information for details of calculating $\theta_U$). The rate of monomer release or consumption ($d[M]/dt$) can then be calculated from the slopes of the curves ($d\theta_U/dT$), the temperature scan rate ($dT/dt$), and the total concentration of subunits ($C_T$) according to the simple expression[13]

$$\frac{d[M]}{dt} = C_T \frac{d\theta_U}{dT} \frac{dT}{dt} \qquad (1)$$

Our method relies on the fact that the set of curves obtained with different scan rates sample multiple $[M]$ and $d[M]/dt$ values at any given temperature within the transition region (Fig. 1a). These measurements yield the effective reaction orders as follows: If the assembled structure contains $N$ subunits and assembly and disassembly occur with effective reaction orders of $n$ and $m$, respectively, then the rate of monomer formation (or consumption) is given by

$$\frac{d[M]}{dt} = -k_{on}[M]^n + k_{off}N\left(\frac{C_T - [M]}{N}\right)^m \qquad (2)$$

When the degree of hysteresis is small, melting and annealing curves lie close together, both terms on the right-hand side of Eq. (2) are similar in magnitude, and the values of $N$, $n$, and $m$ must be known a priori in order to extract values of $k_{on}$ and $k_{off}$[18,19]. The situation is considerably simpler when the degree of hysteresis is large. In this case, the first term dominates during the annealing scan and

$$\frac{d[M]}{dt} \approx -k_{on}[M]^n \qquad (3)$$

A plot of $\log(-d[M]/dt)$ vs. $\log([M])$ is therefore linear with a slope of $n$ and $y$-intercept of $\log(k_{on})$. During the melting scan,

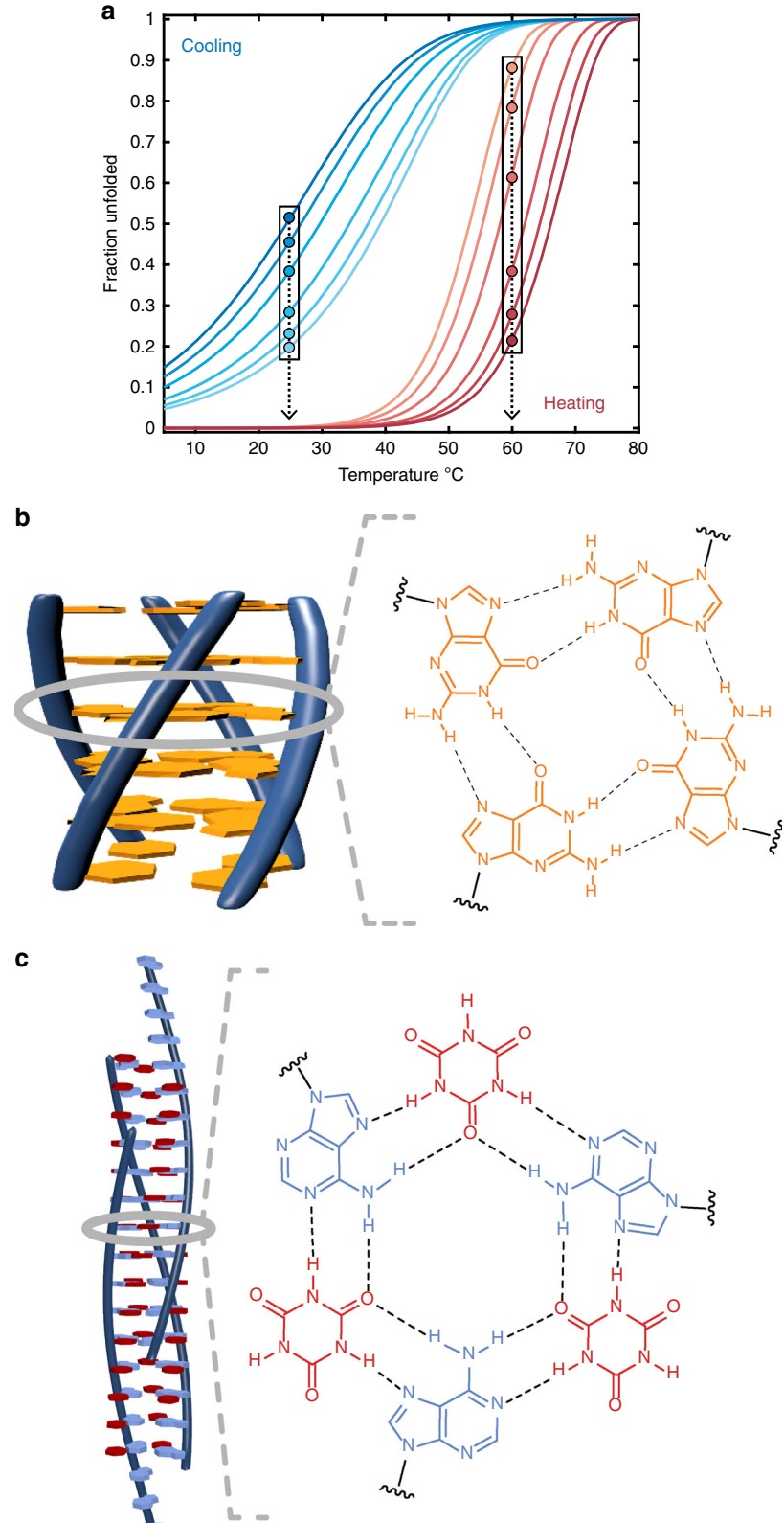

**Fig. 1** Supramolecular assemblies and model-free analysis of multi-scan rate TH datasets. **a** A multi-scan rate TH dataset for generating a 3D supramolecular assembly map. The dashed arrows indicate temperature slices at which monomer concentrations and reaction rates are calculated at each temperature scan rate (colored circles in the black boxes). Light to dark blue and orange to dark red indicate slow to fast scan rates, respectively. **b** Tetrameric GQ structure formed by TG$_4$T. The tetramolecular structure (left) contains stacked, Hoogsteen-hydrogen bonded G-tetrads (right). **c** Fiber structure formed by co-assembly of CA and poly(A) strands (left). CA brings about the growth of nanofibers from poly(A) strands by participating in hexameric rosette arrays with adenine residues (right)

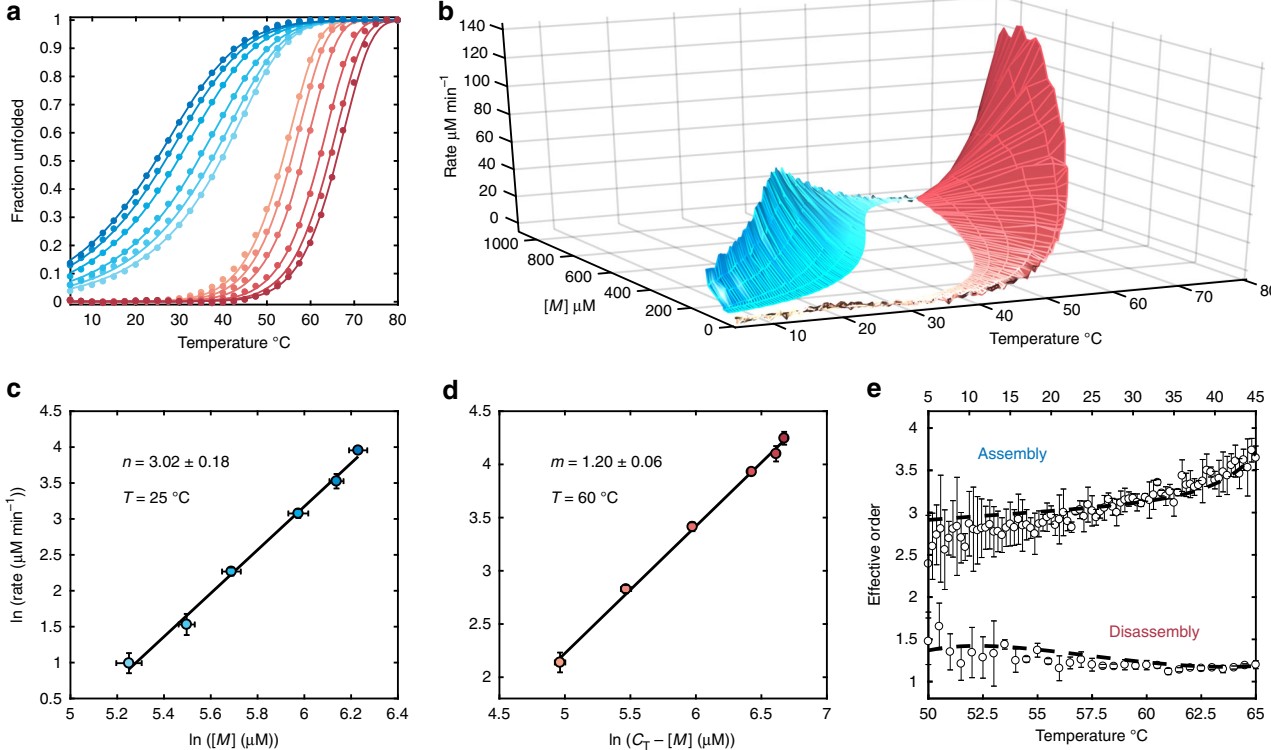

**Fig. 2** Mapping the energy landscape of TG$_4$T assembly by TH. **a** Multi-scan rate fraction unfolded TH profiles for TG$_4$T assembly and disassembly. Colored points and lines are experimental data and the globally-fitted step-wise monomer association model, respectively. Only every fifth experimental point is shown for clarity. **b** 3D temperature-concentration-rate supramolecular assembly map calculated from the experimental data in **a**. The monomer reaction rates (shown as absolute values) increase with faster scanning. **c** Isothermal slice from **b** at 25 °C plotted as ln(assembly rate) vs. ln([$M$]). The effective assembly order $n$ is the slope of the line. **d** Isothermal slice from **b** at 60 °C plotted as ln(disassembly rate) vs. ln($C_T − [M]$). The effective disassembly order $m$ is the slope of the line. **e** Effective TG$_4$T assembly and disassembly reaction orders from model-free analysis of the surface in **b** as a function of temperature through the annealing and melting transitions. The top and bottom temperature axes are for assembly and disassembly, respectively. White circles and dashed black lines correspond to effective orders from experimental data and the globally fit step-wise monomer association model, respectively. In **a–d**, light to dark blue and orange to dark red corresponds to slow to fast annealing and melting scan rates, respectively. In **c–e**, error bars are the standard deviations of the values obtained from analysis of three replicate TH datasets

the second term dominates and

$$\frac{d[M]}{dt} \approx k_{off} N \left( \frac{C_T - [M]}{N} \right)^m \qquad (4)$$

A plot of log(d[$M$]/d$t$) vs. log($C_T − [M]$) is therefore linear with a slope of $m$ and y-intercept of $(1 − m)\log(N) + \log(k_{off})$. The values of $n$ and $m$ thus obtained provide model-free estimates of the reaction orders at each temperature throughout the transitions, while $k_{on}$ and $k_{off}$ are essentially phenomenological constants describing the rate of the reaction. The magnitude of hysteresis required for these approximations may be judged by comparing the slopes of the melting and annealing curves at any given temperature. We consider a ratio of slopes of roughly threefold or more between the slowest annealing and melting scan rates in the middle of the annealing transition to be sufficient, although a more rigorous examination of this approximation can be achieved by numerical simulation. Fortunately, the degree of hysteresis can be tuned by changing the scan rates and $C_T$. Increasing the scan rate and lowering the total concentration of subunits tend to increase hysteresis (we consider shifts in the apparent $T_m$ of ≥1 °C to be suitable). Therefore, this approach can be applied to a wide variety of

supramolecular assembly processes with careful selection of the experimental conditions.

**Assembly of a tetrameric DNA guanine quadruplex**. To test the TH method, we applied it to the well-studied tetrameric deoxyribonucleic acid (DNA) guanine quadruplex (GQ) TG$_4$T (Fig. 1b), which is believed to fold via a pathway involving small populations of partly assembled intermediates[7]. We used spectroscopic absorbance measurements to determine the fraction of unfolded DNA strands as the temperature was raised and lowered at rates varying from 0.2 to 2 °C min$^{-1}$. (Fig. 2a, Supplementary Figs. 1, 2, 3a, b and 4, and Supplementary Methods for details of baseline and temperature correction). Equation (1) was then applied to map assembly (cooling) and disassembly (heating) rates as a function of temperature and monomer concentration (Fig. 2b). Slices through this landscape perpendicular to the temperature axis yield reaction rates as a function of [$M$] at constant temperature. Log-log plots were constructed (Fig. 2c, d) yielding linear correlations, as predicted by Eqs. (3) and (4). The level of agreement is remarkable, as each point in the graph is obtained from a separate melt with a different temperature scan rate, giving us confidence in the analysis that follows. The slopes of the plots correspond to the effective reaction orders of assembly and disassembly sampled

as a function of temperature (Fig. 2e). Assembly reaction orders, $n$, are roughly 2.75 at 5 °C and gradually rise to about 3.5 at 45 °C, while disassembly orders, $m$, are steady near 1.2 from 65 down to ~55 °C. Perfectly simultaneous assembly of all four strands would produce assembly reaction orders of exactly 4. These results therefore imply that the dominant energy barriers for assembly of this GQ are crossed by intermediates with fewer than four strands. A similar temperature-dependent increase in $n$ from roughly 3 to 4 was previously reported[7], which is notable as that study employed a series of variable-concentration isothermal folding reactions monitored by nuclear magnetic resonance (NMR) spectroscopy. Our results are thus in excellent agreement with those obtained from a completely orthogonal methodology. Furthermore, it must be emphasized that the total experiment time of the previous study was on the order of months and sampled only six temperatures, while our data set was obtained in 24 h and sampled orders at 80 different temperatures.

We suspected that the observed variation of the assembly reaction order is likely due to temperature-dependent shifts in the energetic barriers along the assembly pathway. To test whether this hypothesis is consistent with the data, we simulated TH curves using kinetic models with explicit assembly intermediates and examined the extent to which they could reproduce the experimental datasets and effective reaction orders (Supplementary Fig. 5, see Supplementary Methods section). A one-step *monomer ↔ tetramer* model gave very poor agreement with the TH data (Supplementary Figs. 5a and 6a), as expected from the extracted experimental values of $n < 4$. The assembly of $TG_4T$ and other similar tetramolecular GQs has been proposed to follow either step-wise (*monomer ↔ dimer ↔ trimer ↔ tetramer*)[7] or dimer-of-dimers type (*monomer ↔ dimer ↔ tetramer*)[20] mechanisms (Supplementary Fig. 5b, c). Both models gave generally good agreement with the raw data and their corresponding intermediate populations never reached more than ~5%, consistent with the effectively two-state assembly previously observed for $TG_4T$ (Supplementary Fig. 6b, c). However, the step-wise model fit substantially better than the dimer-of-dimers model (roughly 1.4-fold in terms of residual sum-of-squares). According to the Akaike Information Criterion[21], the relative likelihood of the dimer-of-dimers model being correct is <0.01% and therefore the step-wise model is preferred. The simulated melting/annealing curves and reaction orders for the step-wise model are shown in (Fig. 2a, e) and agree closely with both experimental TH data and extracted reaction orders. The extracted rate constants are physically reasonable: the dimer intermediate is highly unstable at 45 °C, with an equilibrium dissociation constant, $K_D$, of ~17 M, and forms very slowly with a kinetic association constant of only 300 $M^{-1}$ $min^{-1}$. Addition of a third strand is more favorable and occurs more rapidly, with a $K_D$ of ~2 μM and association rate constant of ~2 × $10^5$ $M^{-1}$ $min^{-1}$, similar to the association kinetics of intermolecular triplex DNA[22]. A simple and realistic physical kinetic model is thus consistent with the temperature dependence of the reaction orders obtained from our model-free analysis, and the values of the extracted step-wise rate constants given in Supplementary Table 1 provide quantitative insight into the nature of the assembly pathway.

**Co-polymerization of polydeoxyadenosine and cyanuric acid**. It was recently discovered that short polydeoxyadenosine (poly(A)) chains co-assemble with cyanuric acid (CA) to form long fibers[23]. A cross-section of the proposed structure (Fig. 1c) shows three adenine residues from different DNA strands hydrogen bonded to three CA molecules and forming stacked, planar, hexameric

rosettes perpendicular to the fiber axis. This system provides a different type of challenge for the TH method than does the assembly of the tetrameric GQ. Rather than identifying specific intermediates formed on route to a well-defined final structure, characterizing poly(A) fiber formation corresponds to elucidating the supramolecular polymerization mechanism i.e., determining how individual poly(A) chains initiate and add to indefinitely growing fibers. We performed TH measurements of poly(A) fiber formation in the presence of excess CA. Heating scans produced identical curves regardless of the scan rate, indicating that dissociation occurs too rapidly at these temperatures to characterize using this method. In contrast, the cooling scans showed a pronounced scan rate dependence (Fig. 3a, Supplementary Fig. 3c, d) and were subjected to further analysis. The rate of unfolded poly(A) consumption was calculated as a function of temperature and concentration (Fig. 3b) and the resulting log-log plots (Fig. 3c) were linear. The apparent reaction orders for assembly were calculated as the slopes of the plots, yielding values close to 3 (Fig. 3d). It must be noted that apparent reaction orders for polymerization reactions do not reflect a single rate-limiting barrier as monomers are consumed by adding to an ensemble of nascent fibers of different lengths[8]. Effective reaction orders can nevertheless yield mechanistic insight and can be related to the molecularity of the nucleus[8,9], as discussed below (see Supplementary Methods).

We tested whether a simple kinetic model could account for the temperature-dependent annealing curves and reaction orders by fitting the Goldstein-Stryer assembly model[8] directly to the experimental data (Supplementary Fig. 7). This model explicitly tracks the populations of all oligomers up to a certain number of monomer units (100 in Fig. 3a, see Supplementary Methods), while populations of longer fibers were accounted for using the approximation of Korevaar et al.[10,24]. Association and dissociation of monomers and short oligomers less than the critical nucleus size, $s$ (where $s$ is the number of poly(A) strands in our case), were described by the nucleation rate constants $k_{n+}$ and $k_{n-}$, respectively, while oligomers larger than $s$ were described with the elongation rate constants $k_{e+}$ and $k_{e-}$. For the critical nucleus itself, the association rate constant was taken as $k_{e+}$, while dissociation was taken as $k_{n-}$. We held the forward rate constants as equal, $k_{n+} = k_{e+}$, a simplification previously applied in other systems[8–10,25], and allowed the assembly activation enthalpies to vary with temperature (i.e., $\Delta C_p \neq 0$)[26]. We applied the model and systematically varied the value of $s$ from 1 to 7 to find the optimal nucleus size (Supplementary Fig. 8). Note that a nucleus of one in this case corresponds to non-cooperative assembly, where monomers bind as strongly to other monomers as they do to oligomers of any other length. Excellent fits were obtained with nucleus sizes of 2–4, with substantial worsening of the fit quality below or above these nucleus sizes. While nucleus sizes of 2–4 are all physically realistic for poly(A) fiber assembly, the best fit was obtained with a nucleus of three and therefore this is our preferred nucleus size. The forward rate constants of $k_{n+} = k_{e+} \approx 7 \times 10^4$ $M^{-1}$ $min^{-1}$ (Supplementary Table 2) are somewhat slower than those of duplex DNA, which is to be expected, given that each poly(A) strand joining the growing fiber must simultaneously organize a column of CA molecules along the interface. The trimeric nucleus is relatively unstable at 25 °C, with $K_D \approx 100$ μM, compared to a total poly(A) concentration of 50 μM, meaning that it is never more populated than the monomeric state under these conditions. The dissociation equilibrium constant for each subsequent poly(A) strand is much more favourable ($K_D \approx 1$ μM), meaning that fibers spontaneously elongate at poly(A) concentrations above this value. Thus, the full set of TH data for poly(A) fiber assembly

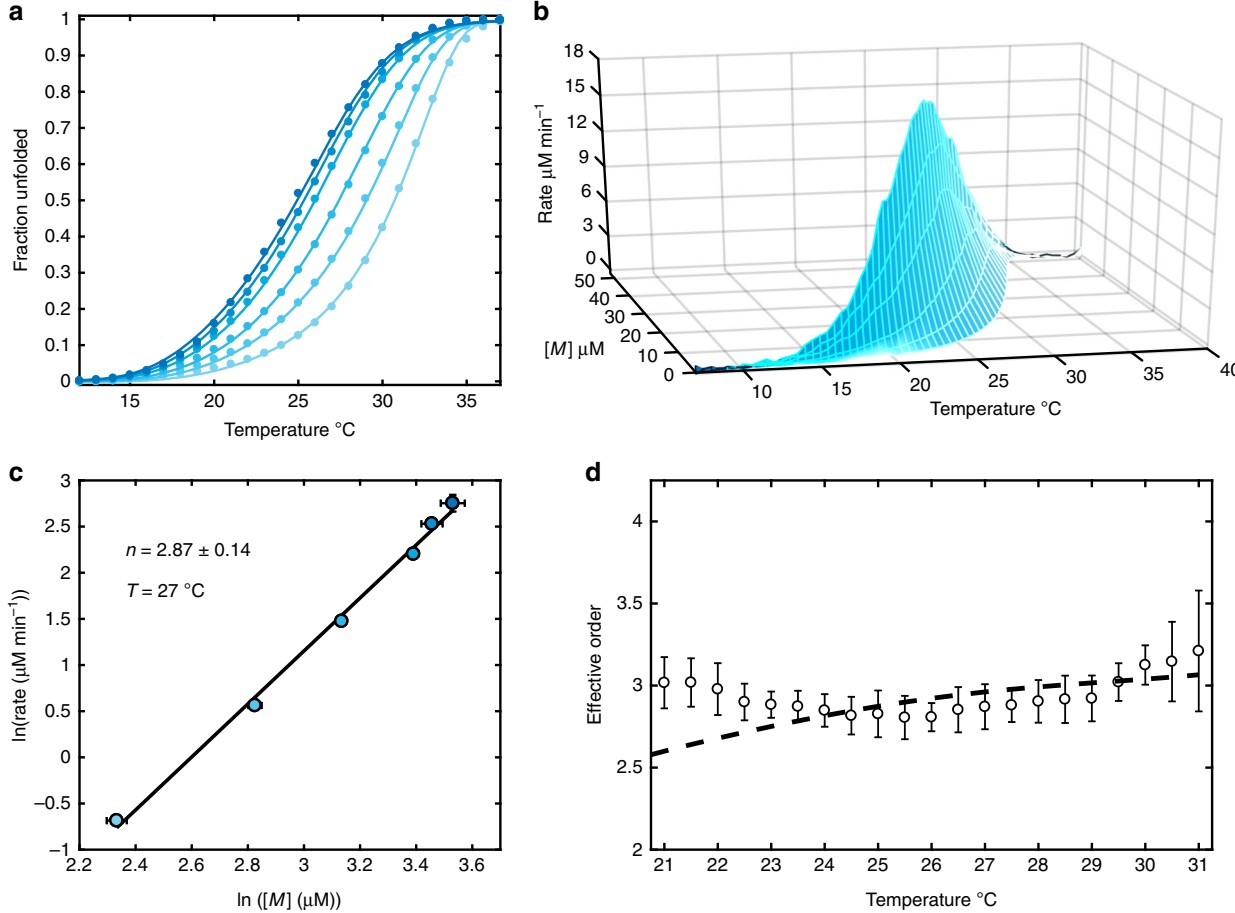

**Fig. 3** Mapping the energy landscape of poly(A) fiber assembly by TH. **a** Fraction unfolded TH profiles for poly(A) fiber assembly as a function of temperature scan rate. Colored points are the experimental data and colored lines correspond to the globally fit Goldstein-Stryer model for cooperative supramolecular assembly assuming a nucleus size of 3. Only every second experimental point is shown for clarity. **b** 3D supramolecular assembly map for poly(A) fiber assembly. **c** Model-free analysis of the map in **b** at 27 °C. The effective assembly order $n$ is the slope of the line. **d** Effective poly(A) fiber assembly orders as a function of temperature through the annealing transition. White circles and dashed black lines are the effective orders obtained from model-free analysis of the experimental and globally fitted data, respectively. In **a**–**c**, light to dark blue corresponds to slowest and fastest annealing scan rates respectively. In **d**, error bars for the experimental points were taken as the standard deviation of the values from analysis of three replicate TH experiments

agrees quantitatively with a simple, realistic, model of supramolecular assembly.

## Discussion

We have shown that a simple analysis of multiple-scan rate TH data yields reaction orders for supramolecular assembly over a broad range of temperatures. This approach is model-free in the sense that no assumptions regarding the populations and inter-conversion rates of partly-assembled intermediate forms are necessary. A multi-scan rate TH dataset can be collected in a few hours with a small amount of material and the extraction of reaction orders is straightforward and can be achieved with standard spreadsheet software. This protocol thus brings kinetic information into easy reach at a level of detail that is not readily obtainable from existing methods.

Likely due to the current scarcity of reaction order data for supramolecular assembly, there has not been much discussion of how values of $n$ and $m$ relate to the underlying pathways. It is therefore useful to examine in more detail how the model-free reaction orders extracted for the tetrameric GQ and poly(A)

fibers relate to the energy surfaces predicted by direct model fitting to the TH curves. In the case of GQ assembly, the experimental values of $n$ are approximately 2.75 at 5 °C, rising to over 3.5 at 45 °C. The reaction energy diagrams predicted by the step-wise model for 5 and 45 °C are shown in Fig. 4a. The heights of the energy barriers correspond directly to the transition probabilities, where larger barriers indicate fewer molecules traversing the barrier in a given direction per unit time. At 5 °C, the first (*monomer* ↔ *dimer*) barrier is larger than the second (*dimer* ↔ *trimer*) and third (*trimer* ↔ *tetramer*) barriers, and the experimental reaction order (2.75) is closer to the molecularity of first transition state (2) than it is to that of the third (4). At 45 °C, the second and third barriers are slightly higher than the first and the observed reaction order (3.5) moves closer to the molecularity of the third transition state. We note that differences in effective order are also somewhat influenced by differences in monomer concentration at different temperatures, but changing barrier heights are the main factor (see Supplementary Methods, Supplementary Fig. 9). In the case of disassembly reaction orders, $m$, the values are all ~1, since the dominant barrier is located at the *tetramer* → *trimer* transition and the corresponding transition

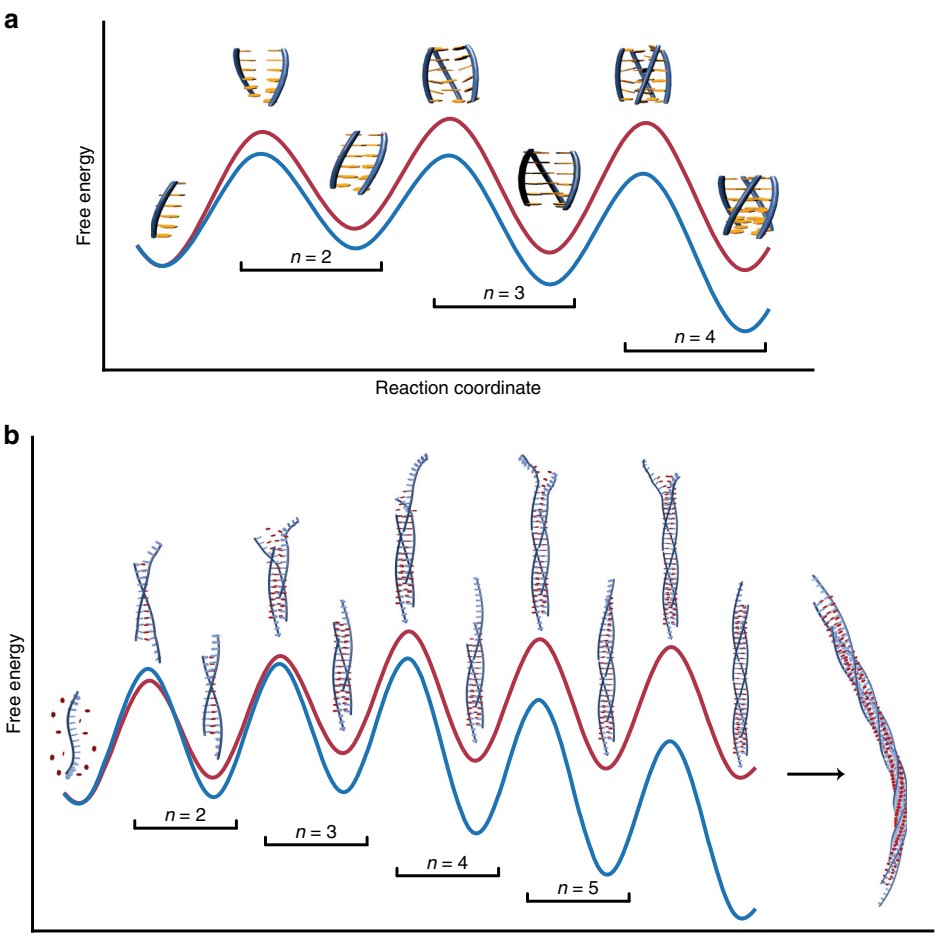

**Fig. 4** Quantitative free energy diagrams for supramolecular assembly by TH. **a** $TG_4T$ assembly at 45 (red) and 5 °C (blue). At 5 °C (experimental $n \approx$ 2.75), the dimer barrier dominates. The trimer and tetramer barriers become dominant at 45 °C (experimental $n \approx 3.5$). **b** Poly(A) fiber formation at 30 (red) and 20 °C (blue) assuming a nucleus size of 3 poly(A) strands. The reaction coordinate is truncated at the hexamer. Longer fibers form from stepwise association of monomers as indicated by the black arrow. In both panels, $n$ values indicating assembly molecularities are given above the black brackets

state has the same molecularity as the fully folded GQ. The temperature-dependent effective reaction orders thus reveal detailed information on the locations and sizes of energetic barriers along the reaction pathway.

For poly(A) fiber formation, the reaction energy diagram predicted by the Goldstein-Stryer assembly model with a nucleus size of 3 is shown in Fig. 4b. The least stable state is the trimeric nucleus, while the addition of each subsequent poly(A) chain produces a successively more stable oligomer. This matches the proposed structure of the fiber, which requires a minimum of three strands to complete the rosette arrangement, and suggests that the addition of a fourth strand effectively stabilizes the nascent fiber in an arrangement that is primed to bind to additional chains. We note that, while in this case, the effective order of the reaction ($\approx 3$) matches the molecularity of the trimeric nucleus, this relationship does not necessarily hold for polymerization reactions in general. Monomers are consumed at each step of the assembly process and elongation continues indefinitely so no single energy barrier is completely rate-determining. Nevertheless, the effective order of a polymerization reaction can be quantitatively interpreted in terms of the fluxes of the individual steps. The net rate at which the $N$-mer oligomer binds

monomers to produce $(N+1)$-mers is given by

$$\Phi_N = k_{on}c_1c_N - k_{off}c_{N+1} \qquad (5)$$

where $c_N$ is the concentration of the $N$-mer, and $k_{on}$ and $k_{off}$ are the appropriate association and dissociation rate constants. The total rate of monomer consumption, $R$, is then

$$R = -\frac{\partial c_1}{\partial t} = \sum_{N=1}^{\infty} \Phi_N \qquad (6)$$

It can be shown (see Supplementary Methods) that the effective order, $n$, of monomer consumption is given by

$$n = \frac{\partial \ln\{R\}}{\partial \ln\{c_1\}} = \sum_{N=1}^{\infty} \frac{\Phi_N}{R} \frac{\partial \ln\{\Phi_N\}}{\partial \ln\{c_1\}} \qquad (7)$$

In other words, the effective order of the polymerization reaction is given by the weighted average of the orders of the individual fluxes, where the weight of each term is simply the relative contribution of the individual flux to the total rate, $\Phi_N/R$. According to the Goldstein-Stryer model applied to poly(A) fiber assembly, the major fluxes for monomer consumption involve

dimers up to about 10-mers and have orders ranging from about 2 to 5, with a weighted average of roughly 3. The order of each flux $\left(\frac{\partial \ln\{\Phi_N\}}{\partial \ln\{c_1\}}\right)$ is largely governed by how the steady-state concentration of the $N$-mer varies with monomer concentration, and on the magnitude of the depolymerization rate $(k_{off}c_{N+1})$ (see Supplementary Methods). We have simulated sequential polymerization reactions according to the Goldstein-Stryer model and find that, without changing rate constants, larger nuclei produce larger effective reaction orders (Supplementary Fig. 10). Thus, the effective reaction orders provide information on the size of the assembly nucleus, $s$. This is particularly true for canonical nucleated assembly, where the concentration of fibers scales as $([M]_{init})^{\frac{s+1}{2}}$ and the rate scales as $([M]_{init})^{\frac{s+3}{2}}$, giving an apparent reaction order of $(s+3)/2$, with respect to the initial monomer concentration, $[M]_{init}$, in isothermal annealing reactions[8]. For the sake of comparison, we have simulated TH data for canonical nucleated assembly. TH experiments are quite different from isothermal annealing reactions since the temperature varies throughout the measurement leading to fiber accumulation that varies with scan rate. Nevertheless, we find empirically similar relationships such that the concentration of fibers scales approximately as $([M])^{\frac{s-1}{2}}$ and the polymerization rate scales as $([M])^{\frac{s+1}{2}}$, giving an apparent reaction order of $(s+1)/2$, where in this case $[M]$ is the actual (instantaneous) monomer concentration (see Supplementary Methods, Supplementary Fig. 11). We note that for poly(A) assembly this empirical relationship would predict an effective order of 2 while we observe orders closer to 3, but poly(A) assembly does not meet the criteria for a canonical nucleated mechanism so the lack of agreement is unsurprising. While we find that effective reaction orders obtained from TH data are information rich and closely linked to the sizes of critical nuclei for self-assembly, the precise relationship is complex and would be an interesting area for further theoretical study.

Our TH-based approach is applicable to many different types of supramolecular self-assembly systems and thus represents a general approach for studying these complex processes. The main requirements are that the reaction is reversible and temperature-controlled, the components are soluble over the entire temperature range[27], and that the degree of self-assembly correlates with a real-time observable such as spectroscopic absorbance or ellipticity. In order to estimate suitable timescales of assembly, we simulated TH datasets for realistic bimolecular association reactions with a range of rates, based on the kinetic parameters we obtained for TG$_4$T. We found that reactions proceeding with apparent rate constants of less than or equal to roughly 1 min$^{-1}$ yielded shifts in the apparent melting temperature of 1 °C or more with accessible temperature scan rates, making this (1 min$^{-1}$) the approximate upper kinetic limit for this technique (Supplementary Fig. 12). Nucleic acids are particularly amenable to this approach, as illustrated by the results presented here. There is growing interest in understanding nucleic acid self-assembly in molecular biology[28] and biotechnology[3], providing many interesting opportunities for application of this method. Furthermore, biological and biomimetic systems such as collagen fibers[18,29], SNARE (Soluble N-ethylmaleimide-sensitive factor Attachment protein REceptor) proteins[30], ganglioside micelles[31], viral capsids[32], peptide amphiphiles[33], elastin-mimetic peptides[34,35], and DNA ribbons[36], along with many others[37] exhibit TH in temperature-driven assembly and represent excellent candidates for TH-based analysis of their assembly mechanisms. In addition, this approach is equally well applicable to non-biological assembly processes, such as rod formation by trisurea disks[38]. Some self-assembling systems exhibit pathway complexity (or pathway selection), following distinct pathways to different kinetically trapped (non-equilibrium) assembled states depending on the reaction

conditions[10,33,39]. Multi-scan rate TH experiments are a potentially powerful way to characterize this class of system and represent an interesting avenue for future research. Importantly, a complete TH dataset can be acquired very rapidly, in as little as a single day, compared to weeks or months for comparable existing methods[7,8]. Interestingly, when melting/annealing kinetics are slow, a TH dataset can be measured in much less time than an equilibrium melting experiment, as there is no need to allow the system to fully equilibrate at each temperature. Extracting the model-free effective reaction orders and rate constants (Eqs. 1–4) is easily accomplished without specialized software. The reaction orders can then be interpreted in terms of the molecularities of the highest energy transition states for discrete assembly, or in terms of nucleus size for polymerization reactions. Subsequently, direct fitting of explicit mechanistic models to the TH data provides detailed insight into the stabilities and interconversion rates of individual assembly intermediates, even those that are very weakly populated and short-lived.

Information on partly-formed intermediates is critical to understanding, and ultimately controlling, macromolecular assembly processes. The growing interest in this challenging problem has led to the development of a variety of biophysical approaches. Many of these focus on direct observation of assembly intermediates. When assembly is extremely slow, intermediates may be sufficiently long-lived for direct structural analysis. For instance, Aβ oligomeric precursors to amyloid fibril formation represent the dominant species after several days of incubation for some variants, and were recently characterized by solid-state NMR and infrared spectroscopy[40]. Alternatively, partly-assembled intermediates can be distinguished from monomers and fully-formed structures by single-molecule methods. For instance, cryo-electron microscopy and atomic force microscopy were used to identify and characterize partly assembled viral capsids on the basis of shape, and to track their abundance as a function of time[41]. Single-molecule microscopy and spectroscopy can identify individual intermediates on the bases of Förster resonance energy transfer intensity or diffusion rates[42], and yield information on their populations and lifetimes[43]. NMR spectroscopy is highly sensitive to millisecond association kinetics and was recently used to dissect a dimer-of-dimers association pathway, giving overall kinetic parameters and identifying interaction surfaces[44]. In contrast with these techniques which require costly specialized equipment, extensive user expertize, and lengthy analysis, multi-scan rate TH analysis can be performed with only a thermally-controlled spectrophotometer and rapidly yields quantitative information on the assembly process in a straightforward manner, in the form of temperature-dependent reaction orders. Similar data are not readily accessible via existing approaches and are highly complementary to those of the more specialized techniques listed above. TH experiments focus on the relative sizes of the kinetic barriers along the assembly pathway rather than on the properties of individual assembly intermediates which may or may not be on-pathway or kinetically relevant. Used in combination with the techniques mentioned above, they can help to identify which of the intermediates are involved in the rate limiting step(s) of assembly. Furthermore, given the simplicity, speed, and low cost of multi-scan rate TH analysis, it is highly suitable as an initial screening method for optimizing samples and assembly conditions for more detailed study. The approach laid out here thus represents a powerful tool for better understanding supramolecular assembly.

## Methods

**Acquisition of TH multi-scan rate datasets.** Desalted d(A$_{15}$) and d(TG$_4$T) oligonucleotides were purchased from Integrated DNA Technologies (IDT). UV-Vis absorbance studies were performed using a 1 mm path length quartz cuvette on a Jasco-810 spectropolarimeter equipped with a Peltier temperature control unit and a water recirculator. For studies on d(TG$_4$T) assembly, samples contained 1 mM d(TG$_4$T) in 1xNaCaco buffer. UV-Vis absorbance was monitored at 295 nm for annealing and melting over the 0 to 85 °C temperature range at different scan rates

(0.2, 0.3, 0.5, 1, 1.5, and 2 °C min$^{-1}$), using a 10 min equilibration time at max/min temperatures. For studies on CA-mediated poly(A) assemblies, samples contained 50 μM d(A$_{15}$) and 15 mM CA in 1xAcMg pH 4.5. UV-Vis absorbances were monitored at 252 nm for annealing and melting over the 2 to 65 °C temperature range at different scan rates (0.2, 0.5, 1, 2, 3, and 4 °C min$^{-1}$), using a 5 min equilibration time at max/min temperatures. A layer of silicon oil was applied on top of the sample solutions to minimize evaporation. A stream of nitrogen gas was supplied to the sample chamber to prevent condensation on the cuvette. Curves were obtained in triplicate.

**Code availability**. The MATLAB code for performing the analyses herein is available from the corresponding author upon request.

**Data availability**. The data for performing the analyses herein are available from the corresponding author upon request.

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

## Acknowledgements

The authors thank Dr. Andrea Greschner for the graphic representations of TG$_4$T and CA-mediated poly(A) assembly structures. R.W.H. was supported by an NSERC CRE-ATE Training Program in Bionanomachines scholarship. N.A. and H.F.S. would like to thank NSERC, CIHR, CFI, and the Canada Research Chairs Program for funding. H.F.S. is a Cottrell Scholar of the Research Corporation. A.K.M. was supported by NSERC (grant number 327028–09).

## Author contributions

A.K.M. and H.F.S. conceived the study. N.A. acquired all experimental data. R.W.H. performed all data analysis, and interpreted the results with A.K.M. R.W.H. and A.K.M. co-wrote the manuscript.

## Additional information

**Competing interests:** The authors declare no competing interests.

