## [Peer Review File · Nature Communications]

Reviewers' comments:

Reviewer #1 (Remarks to the Author):

In an excellent manuscript, the authors introduce the spectroscopic thermal hysteresis method for a simple but highly effective manner to get detailed insight into the mechanism of supramolecular assembly processes. The group of this reviewer thought about the approach but never did it and – not surprisingly – this reviewer likes the manuscript very much. It is a very strong approach and worked out in great detail. I like to strongly recommend the manuscript for Nature Communications. Just two suggestions for minor revision.

- 1) The TH method is used in inorganic nanoparticle assembly and maybe it can be added.
- 2) Figure 4b is not so clear the trimer as the nucleus is clear, but what are the dots around the poly(A) and at $n=3$?

Reviewer #2 (Remarks to the Author):

Mittermaier and co-workers presents, in an elegant way, a thermal hysteresis for mapping the energy landscapes of supramolecular assembly. They have shown spectroscopic thermal hysteresis (TH), a technique widely used to measure two-state folding and unfolding rates of biomolecules, as a new tool for further elucidation of complex supramolecular assembly pathways. Herein, what they are proposing is a novel method that, in contrast with NMR, IR, cryo-TEM, AFM, FRET, can be performed with a standard (and low cost) thermally-controlled spectrophotometer, giving straightforward temperature-dependent reaction orders and thus information on the assembly process.

If we consider the fact that all mentioned spectroscopy techniques require, to achieve consistent data, different concentrations and temperatures experiments, high levels of cost and time, spectroscopic thermal hysteresis is surely of interest for the community. I highly recommend publication of this manuscript after addressing the following points.

The authors should stress some points:

1. Page 4, line 1. 'The temperature scan rate is chosen to be rapid compared to the length of time needed for the system to relax to equilibrium, such that both folding and unfolding occur out of equilibrium.' Could the authors indicate the ranges of k_{on} and k_{off} that are available with the TH method considering the available heating/cooling rates (in the order of 0.2 ... 2 deg / min)? This will make it easier for the general reader to assess whether TH is applicable for their system.
2. More generally, the two systems studied in the paper can reach 2 plateaus, i.e., fully assembled and fully disassembled under the conditions used. In many systems, it is very hard to obtain both plateaus. Is this a strict requirement for TH, or is there no need? I can imagine that θ_u can only be determined accurately in case both plateau values are known.
3. As a small practical point, how crucial is having a T-probe directly in solution vs. monitoring the T of the holder. How large are the deviations if the most standard spectrometer (without solution T-probe) is used?
4. One thing the authors could add is that heating/cooling is not as generally applicable for all systems (especially in water, due to entropic effects, LCST or UCST behavior). For sure in artificial supramolecular polymerization (in organic solvents) this approach would be very interesting.

5. In Figure S5 the authors start fitting with nucleus size 1. What is the physical meaning of this? The monomer undergoes an internal activation?

6. The authors will broaden the audience by including the common terminology of “pathway complexity” or “pathway selection” used in artificial supramolecular systems in the abstract, and citing recent tutorial reviews on the topic e.g., <http://dx.doi.org/10.1039/C7CS00121E>.

Overall this work is very exciting and will surely be adapted by the protein/amyloid and supramolecular chemistry communities.

Reviewer #3 (Remarks to the Author):

The authors describe a method to extract details of assembly pathways from thermal profiles of assembly and disassembly. They demonstrate the utility of the method using a well-characterised G-quadruplex system and a less-well characterised polymerisation of polyA. They emphasize that method is rapid and can be applied in a broad range of systems.

The manuscript is very well written and the claims are supported by the data.

Thermal profiles show hysteresis when the temperature ramp is sufficiently fast that either assembly or disassembly (or both) are out of equilibrium. If this condition is met then effective reaction order and effective rates can be extracted. I suspect that disassembly is too fast in most systems to be amenable to this analysis (as is the case with polyA polymerisation) but I do not see this as a major limitation of the technique. I am also concerned that the method might not be able to discriminate between different assembly models for more complex assembly pathways (but of course we will have to wait and see).

This is a useful method. Thermal profiles are often used in the literature to make qualitative statements about assembly so the quantitative description provided in this manuscript is both novel and welcome. As the authors indicate in the discussion, the connection between reaction order and assembly pathway is not well characterised but could, for example in the case of polymerisation, provide information about the nucleation process. I hope that this manuscript stimulates further research along these lines.

Reviewer 1

1) The TH method is used in inorganic nanoparticle assembly and maybe it can be added.

We thank the reviewer for bringing our attention to the use of TH measurements in nanoparticle studies. Indeed our work could potentially be of interest to this community as well. We have updated the Introduction to read:

Motivated by the need for new methods to efficiently characterize the pathways of supramolecular assembly, we turned to thermal hysteresis (TH), a simple and rapid technique that had previously been used mainly to measure two-state folding and unfolding rates of biomolecules^{13, 14}, as well as the magnetic properties^{15, 16} of inorganic nanoparticles.

2) Figure 4b is not so clear the trimer as the nucleus is clear, but what are the dots around the poly(A) and at n=3?

The dashed box around the trimeric poly(A) transition state was originally placed for visual consistency with Figure 4a (TG₄T). In retrospect, we agree with the reviewer that this is distracting as our discussion of the poly(A) reaction order focuses on nucleus size and the addition of monomers to short fibers of varying lengths, rather than on the molecularity of transition states. We have removed the boxes and instead included labels indicating the molecularities of both transition states and assembly intermediates in both panels. Figure 4 now appears as:

Figure 4. Quantitative free energy diagrams for supramolecular assembly by TH. (a) TG₄T assembly at 45 (red) and 5 (blue) °C. At 5 °C (experimental $n \approx 2.75$), the dimer barrier dominates. The trimer and tetramer barriers become dominant at 45 °C (experimental $n \approx 3.5$). **(b)** Poly(A) fiber formation at 30 (red) and 20 (blue) °C assuming a nucleus size of 3 poly(A) strands. The reaction coordinate is truncated at the hexamer. Longer fibers form from step-wise association of monomers as indicated by the black arrow. **In both panels, n values indicating assembly molecularities are given above the black brackets.**

Reviewer 2

1) Page 4, line 1. ‘The temperature scan rate is chosen to be rapid compared to the length of time needed for the system to relax to equilibrium, such that both folding and unfolding occur out of equilibrium.’ Could the authors indicate the ranges of k_{on} and k_{off} that are available with the TH method considering the available heating/cooling rates (in the order of 0.2 ... 2 deg / min)? This will make it easier for the general reader to assess whether TH is applicable for their system.

The reviewer brings up an important question for assessing when our approach is applicable. In our experience, the minimum shift in apparent melting temperature that gives useful TH

data is about $|\Delta T_m| = 1$ °C. We performed additional simulations to estimate the kinetics of association and dissociation that yield TH of this magnitude. The results are shown in a new Supplementary Figure 10 with simulation details given in the figure legend as follows:

Supplementary Figure 10. Assessing the extent of TH as a function of assembly and disassembly kinetics for a dimeric system $2M \leftrightarrow M_2$. **(a)** $\text{Log}_{10}(|\Delta T_m| = |T_{50\%} - T_m|)$ (where $T_{50\%}$ and T_m are the 50% assembled or disassembled and equilibrium melting temperatures respectively) versus $\text{log}_{10}(k_{ex} \text{ (min}^{-1}) = k_f[M] + k_u)$ at the T_m (i.e. $[M] = 2[M_2]$, $k_f C_T = k_u$, $k_{ex} = (1/2)C_T k_f + C_T k_f = (3/2)C_T k_f$) for 0.2, 0.5, 1, and 2 °C min⁻¹ scan rates. **(b,c)** $\Delta T_m = T_{50\%} - T_m$ versus $\text{log}_{10}(k_{ex})$ at the T_m for **(b)** 0.2, 0.5, **(c)** 1, and 2 °C min⁻¹ scan rates. 1000 noiseless TH datasets were randomly generated using a total monomer concentration $C_T = 10 \mu\text{M}$ and simulation parameters drawn from uniform probability distributions over the intervals [-10, -80] kcal mol⁻¹, [50, 5×10^5] M⁻¹ min⁻¹, [20, 90] kcal mol⁻¹, and [5×10^{-2} , 5×10^{-5}] min⁻¹ for E_f , $k_{f,0}$, E_u , and $k_{u,0}$ respectively. T_m values were calculated according to $T_m = \Delta H / [\Delta S + R \ln(C_T)]$. ΔH and ΔS of association were calculated as $(E_f - E_u)$ and $(\Delta H / T_{ref} + R \ln(k_{f,0} / k_{u,0}))$ respectively. Temperature dependent k_f and k_u values were calculated according to Eq. 13 (above) and T_{ref} was set to 298 K. Simulated annealing and melting data are plotted as filled circles and diamonds respectively. Dark blue to light green symbols indicate slowest to fastest scan rates respectively. Dashed black lines indicate $\Delta T_m = \pm 1$ °C, an approximate lower limit for reliable analysis. Note that kinetics are highly tunable by varying C_T , since $k_{ex} = (3/2)k_f C_T$ at the mid-point of the transition. Reducing the total monomer concentration by a factor of 10 decreases the assembly kinetics by a factor of 10 in the transition region relevant to TH experiments.

The text on page 15 now reads:

In order to estimate suitable timescales of assembly, we simulated TH datasets for realistic bimolecular association reactions with a range of rates, based on the kinetic parameters we obtained for TG₄T. We found that reactions proceeding with apparent rate constants of less than or equal to roughly 1 min⁻¹ yielded shifts in the apparent melting temperature of 1 °C or more with accessible temperature scan rates, making this (1 min⁻¹) the approximate upper kinetic limit for this technique (Supplementary Fig. 10).

2) More generally, the two systems studied in the paper can reach 2 plateaus, i.e., fully assembled and fully disassembled under the conditions used. In many systems, it is very hard to obtain both plateaus. Is this a strict requirement for TH, or is there no need? I can imagine that θ_u can only be determined accurately in case both plateau values are known.

We agree with this reviewer that knowledge of both plateaus is required to calculate θ_u . We have modified the Supplementary Methods to better describe how the plateaus can be

estimated in cases where they are obscured by large amplitude TH. Pages 7-8 of the SI now read:

We note that calculation of $\theta_U(T)$ relies on knowledge of both the assembled and disassembled baselines. This can present a challenge when large amounts of TH are present. For example, assembly of TG₄T reaches ~90% under our experimental conditions with the remainder occurring during the low temperature equilibration period, meaning that low temperature baselines are not directly observed in the annealing scans. However melting scans provide well-defined assembled baselines, which we used as proxies for the assembled baselines of the annealing scans in order to calculate $\theta_U(T)$. The converse approach can be taken when the disassembled baselines cannot be directly observed in melting scans. For systems with extremely slow kinetics, the scan rate, monomer concentration, and solution conditions can be adjusted to promote assembly or disassembly in order to quantify $\theta_U(T)$.

3) As a small practical point, how crucial is having a T-probe directly in solution vs. monitoring the T of the holder. How large are the deviations if the most standard spectrometer (without solution T-probe) is used?

This is indeed a critical issue for obtaining accurate TH data. In order to address this rigorously, we performed a side-by-side comparison of uncorrected thermal melt data, thermal melt data in which we applied the correction described in the SI, and data obtained with a T-probe in solution. Importantly, we find that the corrected data and those obtained with T-probe are essentially indistinguishable, while uncorrected melting and annealing curves show artifactual ΔT_m values of as much as 8-10 °C. Therefore it is essential to either use a T-probe or to perform calibration experiments with which to correct the data, as described here. Pages 6-7 of the SI now read:

We additionally performed a validation of the temperature scan correction approach described above which can serve as a general test of temperature scan corrections obtained for thermal denaturation instruments. The procedure centers on measuring multi-scan rate thermal denaturation and renaturation datasets with a rapidly folding/unfolding sample, which in our case was an intramolecular G-quadruplex formed by the sequence 5'-AGGGTGGGIAGGGTGGGI-3'. For this type of system, the corrected folding and unfolding profiles should coincide (i.e. the TH should be approximately zero) at all tested scan rates. We measured heating and cooling profiles during which the instrument actively controlled the block temperature, while passively monitoring the sample temperature with a probe (Supplementary Figure 4a). Using this calibration, we performed the linear temperature correction described above (Supplementary Figure 4b). Importantly, this collapsed the profiles to their equilibrium positions, as expected for this type of rapidly folding system. The temperature correction was further validated by performing a separate experiment where the sample solution temperature was controlled directly via the integrated temperature probe (Supplementary Figure 4c). These profiles are nearly identical to the temperature-corrected data, demonstrating that our correction approach accurately reproduces the sample temperatures, without the need for an integrated temperature probe. Supplementary Figure 4a, illustrates that either a probe or temperature correction must be used in order to obtain reliable data. Controlling only the block temperature without correcting for differences between block and sample temperatures leads to artifactually large amounts of TH (~8-10

°C), even for a rapidly folding sample where the system is in fact at thermal equilibrium throughout the experiment.

Supplementary Figure 4. Validation of the temperature correction of multi-scan rate thermal denaturation datasets. (a) Absorbance data collected by actively controlling the block temperature. (b) Data from (a) corrected as described in the Supplementary Methods. (c) Absorbance data collected by actively controlling the sample solution with an integrated temperature probe. Importantly, (b) and (c) produce nearly identical results, validating the temperature correction approach described here. The sample consisted of 5 μM of intramolecular G-quadruplex, 5'-AGGGTGGGIAGGGTGGGI-3', in 10 mM lithium phosphate buffer pH 7.0 with 5 mM KCl. Experiments were measured with a Cary 300 UV-Visible spectrophotometer at 295 nm using scan rates of 1, 1.5, 2, 3, and 4 $^{\circ}\text{C min}^{-1}$.

4) One thing the authors could add is that heating/cooling is not as generally applicable for all systems (especially in water, due to entropic effects, LCST or UCST behavior). For sure in artificial supramolecular polymerization (in organic solvents) this approach would be very interesting.

The reviewer has raised the important point that some molecules undergo large temperature-dependent changes in solubility. One of the caveats of the TH approach is that all components of the assembling system are soluble over the temperature range studied experimentally. We have added a note to this effect in the Discussion. The text on page 15 now reads:

Our TH-based approach is applicable to many different types of supramolecular self-assembly systems and thus represents a general approach for studying these complex processes. The main requirements are that the reaction is reversible and temperature-controlled, **the components are soluble over the entire temperature range²⁷**, and that the degree of self-assembly correlates with a real-time observable such as spectroscopic absorbance or ellipticity.

5) In Figure S5 the authors start fitting with nucleus size 1. What is the physical meaning of this? The monomer undergoes an internal activation?

We have clarified the main text on page 10 as follows:

We applied the model and systematically varied the value of s from 1 to 7 to find the optimal nucleus size (Supplementary Fig. 6). **Note that a nucleus of 1 in this case corresponds to non-cooperative assembly, where monomers bind as strongly to other monomers as they do to oligomers of any other length.**

6) The authors will broaden the audience by including the common terminology of “pathway complexity” or “pathway selection” used in artificial supramolecular systems in the abstract, and citing recent tutorial reviews on the topic e.g., <http://dx.doi.org/10.1039/C7CS00121E>.

We agree that our approach could potentially be very useful for studying pathway complexity in artificial supramolecular systems and are grateful for this suggestion. We have modified the abstract to include the suggested keywords and provided more details and a citation of this excellent review in the Discussion. The abstract now includes:

Understanding how biological macromolecules assemble into higher-order structures is critical to explaining their function in living organisms and engineered biomaterials. Transient, partly-structured intermediates are essential in many assembly processes **and pathway selection** but are challenging to characterize.

The text on page 16 now reads:

Some self-assembling systems exhibit pathway complexity (or pathway selection), following distinct pathways to different kinetically trapped (non-equilibrium) assembled states depending on the reaction conditions^{10, 33, 39}. Multi-scan rate TH experiments are a potentially powerful way to characterize this class of system and represent an interesting avenue for future research.

Reviewer 3

No changes suggested.

REVIEWERS' COMMENTS:

Reviewer #2 (Remarks to the Author):

All points have been addressed. Great work.